# The Psychological Impact on Romanian Women Infected with SARS-CoV-2 during Pregnancy

**DOI:** 10.3390/healthcare12090945

**Published:** 2024-05-04

**Authors:** Ruxandra-Gabriela Cigăran, Gheorghe Peltecu, Laura-Mihaela Mustață, Radu Botezatu

**Affiliations:** 1Department of Obstetrics and Gynecology, Filantropia Clinical Hospital, 011171 Bucharest, Romania; ruxandra-gabriela.cigaran@drd.umfcd.ro (R.-G.C.); gheorghe.peltecu@umfcd.ro (G.P.); laura.mustata@gmail.com (L.-M.M.); 2Department of Obstetrics and Gynecology, Carol Davila University of Medicine and Pharmacy, 020021 Bucharest, Romania

**Keywords:** pregnancy, COVID-19, mental health, healthcare

## Abstract

Background: It is well-known that the uncertainty about the COVID-19 pandemic has an indirect negative impact on pregnant women’s mental health, given the fact that pregnant women are more vulnerable emotionally and psychologically than non-pregnant women. The aim of this study was to evaluate the maternal psychological impact on Romanian women who were infected with SARS-CoV-2 during pregnancy and their concerns and to determine which are the best measures to prevent negative outcomes. Methods: A 40-item questionnaire was created for data collection and was shared on social platforms (Facebook and Instagram) and also with obstetric communities between February 2021 and January 2023. Our cross-sectional survey recruited 317 Romanian pregnant women who suffered from SARS-CoV-2 infection. Among general questions about their life and pregnancy during the pandemic, the survey included questions about their SARS-CoV-2 infection during pregnancy, their concerns and how they perceived this period in order to evaluate their emotional status. Results: Of 317 women recruited, 91% of them had a mild to moderate form of COVID-19, and 2% had serious symptoms. Only 9% of women were hospitalized, 4% of women considered that the SARS-CoV-2 infection affected their physical condition to a great extent, and 8% considered to be affected in terms of mental state to a great extent. The main negative feelings of pregnant women during the COVID-19 disease were the fear regarding the possibility of affecting the pregnancy and the concern for their life (51.4%). These increase the risk of developing anxiety or depression. Pregnant women who contracted SARS-CoV-2 infection faced negative feelings, especially those with a severe form of the disease or who recovered with difficulty after the disease. Patients who required hospitalization reported an impairment of the mental state to a great extent and to a very great extent with a frequency of approximately two times and four times higher than non-hospitalized patients, respectively (*p* < 0.05 and *p* < 0.001, respectively). Also, giving birth during SARS-CoV-2 and the difficulty of accessing medical services represented a high level of stress. Also, 47% of patients who reported difficulty accessing medical services during the illness evaluated their mental state significantly less favorably. Conclusion: Preventive measures are essential to minimizing the negative psychological impact of COVID-19 disease during pregnancy among pregnant women. The medical treatment of COVID-19 disease during pregnancy should be prioritized, but emotional and mental health support must also be provided.

## 1. Introduction

The first documented case of acute respiratory disease of SARS-CoV-2 was in China in December 2019, and the World Health Organization (WHO) announced the start of the COVID-19 pandemic on 11 March 2020 [1]. The SARS-CoV-2 pandemic has affected the global economy, all the structures of countries, healthcare services, and overall, the well-being of the world population. Also, the Romanian authorities have adapted to the pandemic situation, and a lot of changes were implemented in order to protect public health [2].

At that moment, there was limited information about the method of transmission, symptoms, risk of developing complications of COVID-19 and treatment [3,4,5]. Even after three years, there are still some unclear aspects. It is considered that vulnerable people, in terms of health, are susceptible to developing severe forms of COVID-19 [3]. Pregnancy, through their changes, makes women vulnerable. For that reason, pregnant women are at risk for severe symptoms because of SARS-CoV-2 infection [3,4,5,6]. 

Worldwide authorities adopted intensive health precautions to decrease the risk of contracting COVID-19. ”Lockdowns”, social distancing, wearing masks and significant changes in healthcare services were considered risk factors for anxiety and mood disorders among pregnant women [4]. Limited information is available related to the pregnancy outcome in SARS-CoV-2 infection and also vertical transmission [5,7]. COVID-19 in pregnancy is sometimes associated with severe complications such as severe pneumonia with ICU admission or even death [5,6,7,8,9,10,11]. Some risk factors were established as being associated with severe symptoms of COVID-19 during pregnancy, including increasing maternal age, high body mass index and maternal comorbidities (gestational diabetes, hypertension) [7,8,9,10,11]. The negative consequences associated with SARS-CoV-2 infection during pregnancy are infrequent [12,13]. Women with severe infections who were hospitalized are considered at risk for negative pregnancy outcomes such as preterm birth, preeclampsia, cesarean or perinatal death [14]. At the same time, most of the studies showed that there is no evidence of vertical transmission [5,6,7,8,14]. 

There are studies that reported that the most common adverse event of women with COVID-19 during pregnancy was preterm delivery [13,15,16,17,18]. Also, low birth weight or C-section delivery was much higher [19,20]. Even if an increased risk of infection for the neonate during vaginal birth is not demonstrated [21], most of the clinical guidelines recommend a C-section as a mode of delivery. C-section incidence increased during pandemic [22]. Maternal death, stillbirth, miscarriage, preeclampsia, fetal growth restriction, coagulopathy and premature rupture of membranes were rare [23]. 

The uncertainty of the pandemic period and the confusion about the effects of SARS-CoV-2 infection during pregnancy had an indirect negative impact on women’s mental health [3,4,5,6,7]. Pregnant women are more vulnerable in terms of emotional and psychological state than non-pregnant women [24]. Therefore, the pandemic changes represent risk factors for pregnant women in terms of psychological distress [14]. There are several reports that during the pandemic, pregnant women experienced anxiety and depression because of restrictions related to pregnancy care, fear of vertical transmission and lack of social/psychological support [23,25]. 

The development of COVID-19 vaccines was considered essential for the reduction of COVID-19-related morbidity and mortality in the general population, including pregnant women [26]. The Romanian authorities recommended vaccination during pregnancy in any trimester and during the lactation period. However, the COVID-19 vaccine acceptance among pregnant women was low because of limited data on safety and efficacy and the flow of misinformation in social media about the teratogenicity of the vaccine [27]. For pregnant women, the decision of COVID-19 vaccination during pregnancy or not was another reason for stress [27].

The aim of this study was to evaluate maternal psychological impact among Romanian women who were infected with SARS-CoV-2 during pregnancy and their concerns. Meanwhile, the more we know about the negative psychological impact of pregnant women infected with SARS-CoV-2, the more we can intervene to limit side effects.

## 2. Materials and Methods

### 2.1. Design

A 40-item questionnaire was created for data collection [28,29,30]. Our survey recruited only Romanian pregnant women who were infected with SARS-CoV-2. The questionnaire was available online in Romanian language. It was created by a team consisting of obstetricians and psychiatrists using Google Forms, and it was shared on Facebook and Instagram and also within obstetric communities between February 2021 and January 2023. 

Data collected were from closed-ended questions or multiple-choice questions. Participants answered questions about their basic demographic information, pregnancy status, life changes during pregnancy and their experience during the COVID-19 pandemic in terms of medical services. Participants were also questioned about their SARS-CoV-2 infection during pregnancy, their concerns and how they perceived this period in order to evaluate their emotional and psychological impact. 

### 2.2. Participants

A number of 317 self-identified women who were diagnosed with SARS-CoV-2 infection during pregnancy were recruited via Facebook and Instagram, including pregnancy-specific groups. The questionnaire link was also shared with medical communities for distribution through their networks of pregnant women. 

### 2.3. Analysis of Data

Data collected were organized in Python 3 using the Pandas module for converting categorical data to numeric data, extracting individual values from multiple-choice questions, and filtering data used in statistical analyses and visualizations. 

To check the association between responses to different questions, contingency tables were constructed, and statistical analyses were performed using exact Fisher test and chi-square (Scipy module).

For numerical data, correlations were calculated using the Pearson method (Scipy), and visualizations were made using Seaborn and Matplotlib modules. Assessment of the separation of clusters of patients according to self-reported level of change in mental status was performed using Principal Component Analysis (PCA) in the Sklearn module. The analysis of the self-assessed symptoms that contributed independently to the alteration of the mental state/stress level was performed using multinomial logistic regression (Statsmodels module).

The separation of the responses of patients with different degrees of self-declared mental impairment was verified with a support vector machine algorithm with a linear kernel, implemented based on the Sklearn module. The data obtained by PCA, together with the level of mental impairment, were divided 50:50 into training and validation sets, and after training, the accuracy of discrimination of the maximum and minimum levels of mental impairment based on the other questions in the questionnaire was calculated.

## 3. Results

### 3.1. General Aspects of Participants

From our group of 317 participants, 79% of women were between 26 and 35 years old, and 50% of women were primiparous (Table 1). Only 13% of the whole group had medical disorders in pregnancy, such as hypertensive disorders, gestational diabetes and fetal growth restriction (Table 1).

Regarding the attitude of the participants during the pandemic, the majority (88%) declared that they respected the rules and restrictions imposed to prevent infection. 

The main source of information about SARS-CoV-2 for the participants was digital media (64.35%). Also, the obstetrician (47.95%) and the general doctor (36.91%) represented a source of information for women from our study about SARS-CoV-2 infection during pregnancy (Figure 1).

Participants declared, to a large extent (66.88%), that the medical information received from all sources suggested a presumed risk for the fetus in the case of infection with SARS-CoV-2 during pregnancy (Figure 2). Also, 23.66% reported fear of developing serious symptoms during COVID-19 disease, requiring hospitalization in the ICU (Figure 2).

Among the symptoms included in the questionnaire for the assessment of stress during the pandemic, the most frequent was fear of the possibility of affecting the pregnancy (69.1%), followed by panic related to the unknown nature of the disease (33.8%), low energy (33.4%) and fear for the safety of one’s own life and relatives (31.2%); the rarest reported symptoms were somatization, loneliness and hopelessness about the future (Figure 3).

### 3.2. Information about Our Participants during SARS-CoV-2 Infection

In our study group, 55% of participants were diagnosed with COVID-19 by testing following the appearance of specific symptoms, and 21% were tested for SARS-CoV-2 infection because they had direct contact with an infected person and developed specific symptoms. Regarding the source of infection, 24% of women did not know where they obtained the infection from, and 24% declared that their partner was the source of infection and, to a lesser extent, other sources were the cause (members of family, friends or work colleagues). 

Most of the participants (42%) were in the second trimester of pregnancy at the time of the disease, 31% in the third trimester and 27% in the first trimester (Table 2). However, 75% of women did not have difficulties accessing medical services during SARS-CoV-2 infection (Table 2). 

In terms of symptoms, 63% of responders had a mild form of COVID-19, 28% had a moderate form and only 2% had a serious form of the disease (Table 2). Only 9% of women were hospitalized.

Among the symptoms included in the questionnaire for the assessment of stress during the SARS-CoV-2 disease, the most frequent was fear for one’s life or the baby’s life (51.4%), followed by anxiety (34.1%), vulnerability (30, 6%) and helplessness (30%); the rarest affective symptoms reported were the feeling of abandonment, the feeling of stigmatization and not accepting the diagnosis (Figure 4).

Almost all participants in the study (92%) continued the pregnancy follow-up protocol investigations after being treated for COVID-19. Only 4% of women reported that they were refused access to medical services because they did not have a recent negative PCR test. Only 5% of women gave birth during COVID-19. 

For 34% of women, the recovery after infection was easy, and for 27% neither difficult nor easy; 11% reported that the recovery was difficult and 2% very difficult (Table 3). 

Also, 34% of women considered that the SARS-CoV-2 infection affected their physical condition to a small extent, 22% of participants felt their physical condition was greatly affected and 4% felt it was affected to a great extent (Table 3). The self-evaluated psychological impact of the COVID-19 disease during pregnancy in pregnant women is observed in Table 3. A total of 31% of them evaluated their mental state as affected to a small extent, and 17% of women estimated a great impact. To a great extent, 8% of women were considered to be affected in terms of mental state (Table 3).

### 3.3. The Analysis of the Results

A good correlation (Pearson R = 0.559, *p* < 0.001) was observed between the assessment of mental status and the assessment of physical status following SARS-CoV-2 infection. Also, self-rated mental status was positively correlated with the difficulty of recovery from the illness (Pearson R = 0.443, *p* < 0.001) and, to a lesser extent, with the perceived severity of the illness (Pearson R = 0.242, *p* < 0.001) (Figure 5).

A number of 78 (25%) patients reported difficulty accessing medical services during the illness, and these patients evaluated their mental state significantly less favorably (47% of patients reported an altered state to a large or very large extent) compared with patients without difficulties in accessing medical services (only 18% of them reported an altered state to a large or very large extent) (Figure 6).

Patients who required hospitalization during the illness (*n* = 27, 9%) reported an impairment of the mental state to a great extent and to a very great extent with a frequency of approximately two times and four times higher than non-hospitalized patients, respectively (*p* < 0.05 and *p* < 0.001, respectively) (Figure 6). 

Another source of stress associated with increased levels of psychological distress was the refusal of admission to medical visits following discharge in the absence of a negative PCR test (*p* < 0.0001), although this problem was only reported by 13 patients (4%) (Figure 6).

Also, the 15 patients who gave birth during the SARS-CoV-2 infection reported considerably higher levels of mental state impairment compared to the general population.

As stated before, among the symptoms included in the questionnaire for the assessment of stress during the SARS-CoV-2 disease, the most frequent was fear for one’s life or the baby’s life (51.4%), followed by anxiety (34.1%), vulnerability (30.6%) and helplessness (30%). Most patients presented between 0 and 6 stress symptoms (where the stress score is calculated as −1 for “my illness did not influence my perception of pregnancy”, “I did not believe in the diagnosis”, “I did not feel it seemed nothing special”, “I encountered no difficulties” and +1 for all other symptoms), and the number of symptoms reported was positively correlated with the impairment of the mental state (Pearson R = 0.57, *p* value < 0.001). In a multinomial logistic regression analysis, to determine the variables that increase the risk of falling into the categories of greatly and very greatly affected mental state after contact with SARS-CoV-2 disease, reporting depression increased the risk 4 times (*p* < 0.05), guilt 3.25 times (*p* < 0.01), anxiety 3.16 times (*p* < 0.01) and helplessness 2.23 times (*p* < 0.05), while the other stress symptoms were not independent risk factors with statistical significance (Figure 7).

In order to identify the origin of the reasons for the alteration of the mental state of some of the pregnant patients who contracted SARS-CoV-2 during pregnancy, we evaluated to what extent the levels of mental impairment can be identified based on other responses. For this purpose, the Principal Components Analysis was carried out starting only from the questions related to access to medical services, only from affective symptoms during pregnancy/pandemic, only from affective symptoms during the SARS-CoV-2 disease from the questions from all these three sets and from all questions, respectively. The first two principal components were used as data to train and validate the separation of responses between extreme cases of mental impairment (not at all and very high) in a support vector machine (SVM) algorithm. 

As can be seen in Figure 8D, the best separation was obtained using all three sets of questions, where cases with no mental state impairment could be identified with 91% accuracy and those with very high impairment with a precision of 86% (AUROC = 0.915). However, individual sets of questions also had good predictive value, with AUROC values of 0.866 for affective symptoms during the pandemic, AUROC of 0.894 for affective symptoms during SARS-CoV-2 infection, and AUROC of 0.903 for access to health care. In conclusion, both affective symptoms during pregnancy/pandemic, affective symptoms during SARS-CoV-2 disease and assessment of access to medical services were highly associated with the perception of mental state impairment, each of these sets of questions being sufficient to be able to discriminate with greater than 85% accuracy between extreme cases of mental impairment (not at all versus very much). Intermediate impairment cases formed intermediate clusters between these extremes, and their separation with good precision would not be possible even on the basis of all questions in the study (Figure 8C).

## 4. Discussion

So far, most of the studies have suggested a negative impact of the COVID-19 pandemic on the general population, especially among vulnerable people, including pregnant women [3,4]. Both direct negative outcomes through infection but also indirect negative impacts concerning the mental health of pregnant women as a consequence of changes in healthcare or social circumstances were described [23,24,25]. 

Even if most of the studies support that the transmission of the virus from mother to baby is unlikely and COVID-19 disease during pregnancy does not necessarily mean severe symptoms, the attitude of the medical staff and patients regarding SARS-CoV-2 infection during pregnancy or during breastfeeding varies, and this implies uncertainty [23]. As we observed in our group, 66.88% of pregnant women declared that the information received suggested that there were risks for the fetus in case of infection with SARS-CoV-2 during pregnancy, and 23.66% that serious symptoms of COVID-19 could appear during pregnancy. The fear of the possibility of affecting the pregnancy (69.1%), the panic related to the unknown nature of the disease (33.8%), low energy (33.4%) and the fear for the safety of one’s own life and that of loved ones (31.2%) were felt frequently by these women. 

In our group, 91% of responders had a mild and moderate form of COVID-19 and 2% had serious symptoms, only 9% of women were hospitalized. A total of 4% of women considered that the SARS-CoV-2 infection affected their physical condition to a great extent. 

The analysis of our data reveals that the difficulty of recovery after the disease had a more negative psychological impact than the symptoms of the disease. Also, patients who required hospitalization declared a high level of stress. Those who required hospitalization during the illness reported an impairment of the mental state to a great and very great extent. 

Much information about COVID-19 disease is still unclear and constantly changing [15]. Many healthcare services have been implemented new medical guidelines and healthcare infrastructure have been changed. Also, the Romanian healthcare system has gone through important changes. An aggressive attitude of the healthcare services in the management of cases during the pandemic has been observed. For example, the WHO recommended continued breastfeeding, rooming or skin-to-skin contact utilizing infection control practices, but few guidelines recommend this practice [1,25]. Because of these unpredictable changes, many countries reported a high frequency of maternal mental health problems [23,25].

However, in our group, 75% of women did not have difficulties accessing medical services during SARS-CoV-2 infection, and 92% of them continued the pregnancy follow-up protocol investigations after treating the disease of COVID-19. To a small extent, they were refused access to medical services because they did not have a recent negative PCR test, and this was an important cause of stress.

The patients who reported the difficulty of accessing medical services during the illness evaluated their mental state significantly less favorably. Also, the patients who gave birth during the SARS-CoV-2 infection presented considerably higher levels of stress.

Among responders, the most declared frequent feeling during SARS-CoV-2 disease was fear for one’s life or the baby’s life (51.4%), followed by anxiety (34.1%), vulnerability (30, 6%) and helplessness (30%). Regarding the self-rated psychological impact of the COVID-19 disease during pregnancy, 17% of women estimated to be affected to a large extent and 8% of them to a very large extent. We observed that self-rated mental status was positively correlated with the difficulty of recovery from the illness.

Both affective symptoms during the pandemic, affective symptoms during SARS-CoV-2 disease and assessment of access to medical services were highly associated with the perception of mental state impairment.

## 5. Conclusions

We already know that the COVID-19 pandemic was associated with a negative psychological impact on pregnant women because of the changes in their social life, new measures implemented in hospitals, the limited information about SARS-CoV-2 infection and pregnancy which create more concerns about their health and pregnancy outcomes.

Our study reveals that pregnant women who contracted SARS-CoV-2 infection faced negative feelings, especially those who develop a severe form of the disease and require hospitalization or who recover with difficulty after the disease. Also, giving birth during SARS-CoV-2 and the difficulty of accessing medical services represented a high level of stress for them. The fear of the possibility of affecting the pregnancy and the fear for their life or the baby’s life were the main negative feelings of pregnant women during the COVID-19 period. All of these mean a higher risk for pregnant women to develop anxiety or depression during pregnancy or postpartum. Considering these findings, preventive measures are important to be taken to minimize the negative impact of COVID-19 disease during pregnancy among pregnant women.

Psychological support during pregnancy is one of the solutions that could be taken into consideration for improving the mental health of pregnant women and for preventing negative outcomes of pregnancy and long-term adverse effects on mothers and babies. The medical treatment of COVID-19 disease during pregnancy should be prioritized, but emotional and mental health support must be provided. Also, the negative feedback because of restrictions and health services changes must be taken into consideration by healthcare providers when preventive measures are established during the pandemic.

## Figures and Tables

**Figure 1 healthcare-12-00945-f001:**
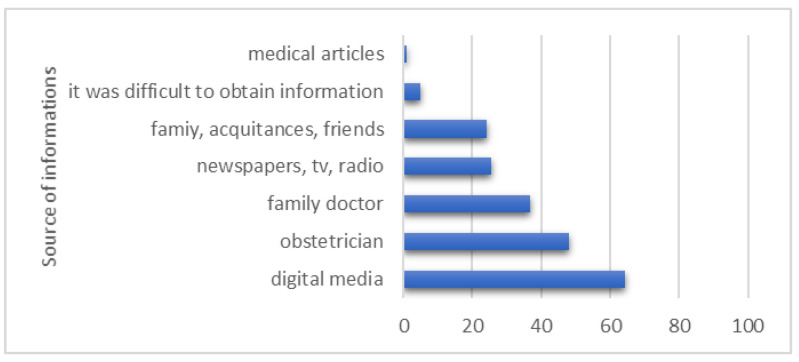
The source of information about SARS-CoV-2 infection during pregnancy.

**Figure 2 healthcare-12-00945-f002:**
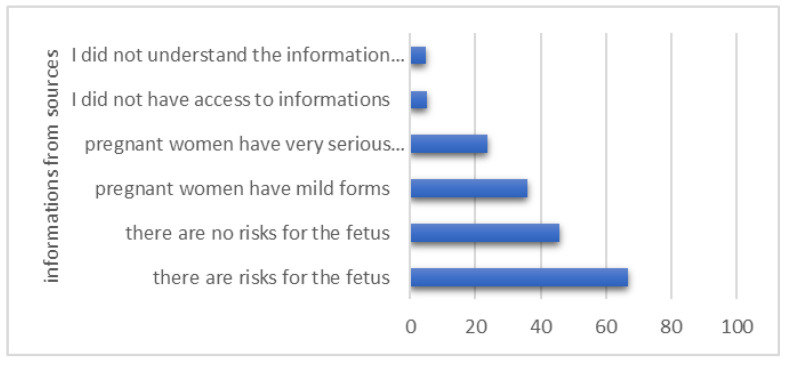
What the sources of information suggested to the participants.

**Figure 3 healthcare-12-00945-f003:**
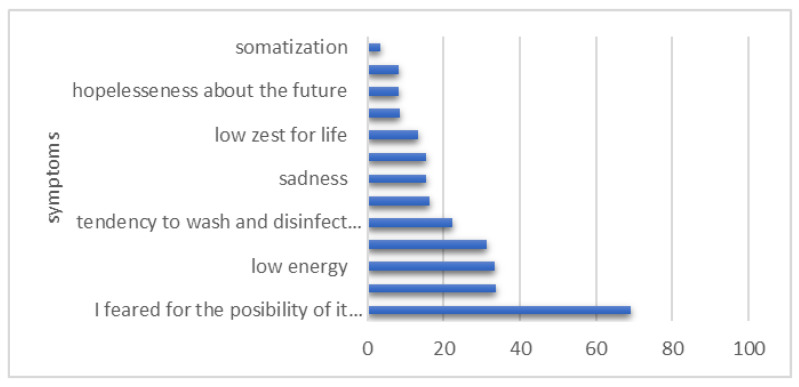
Participants feelings during pandemic.

**Figure 4 healthcare-12-00945-f004:**
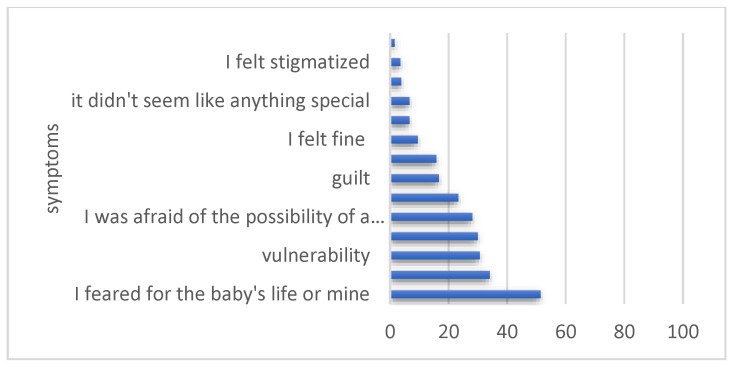
The feelings of participants during COVID-19 disease.

**Figure 5 healthcare-12-00945-f005:**
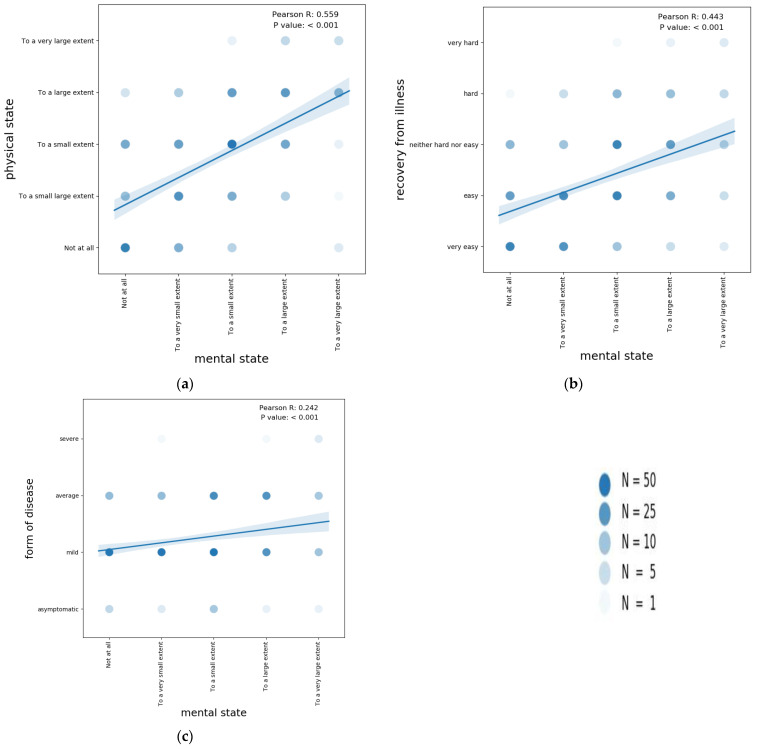
Correlations between mental state and physical state (**a**); Correlations between mental state and recovery from illness (**b**); Correlations between mental state and severity of COVID-19 disease (**c**).The intensity of the color is proportional to the number of patients who had the indicated pair of responses. The graph shows the linear regression line with the 95% confidence interval. Also, the Pearson coefficient and *p*-value of the correlation are indicated.

**Figure 6 healthcare-12-00945-f006:**
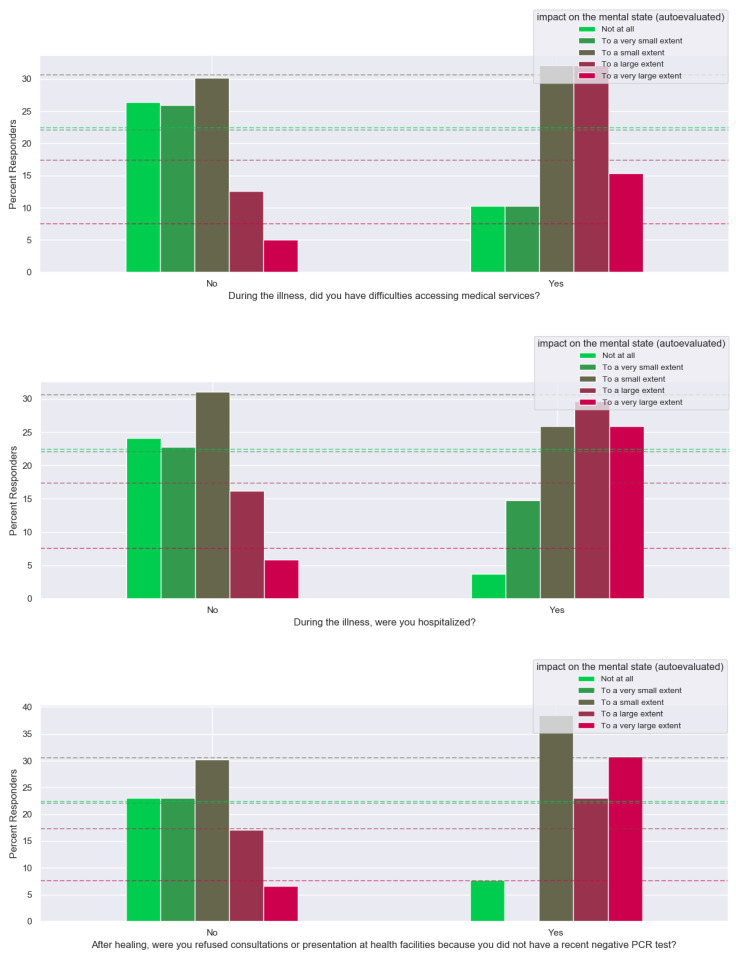
Access to medical services and the impact on the mental state after contact with SARS-CoV-2.

**Figure 7 healthcare-12-00945-f007:**
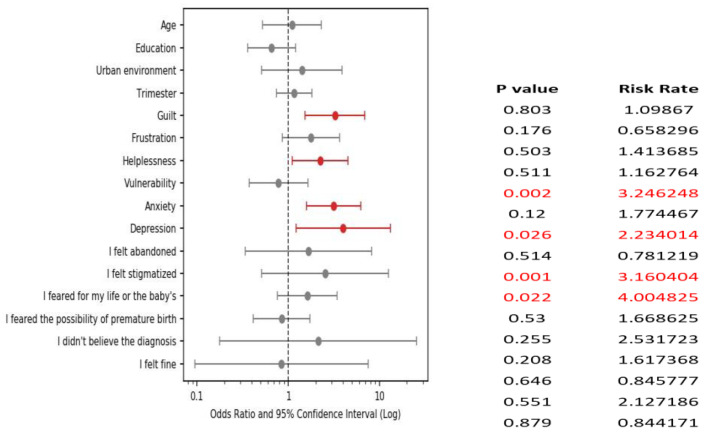
Affective symptoms during the SARS-CoV-2 disease and the impairment of the mental state following the SARS-CoV-2 disease: forest plot representation of multinomial logistic regression analysis. Regarding symptoms “I didn’t believe in the diagnosis” and “it didn’t seem like anything special” the analysis did not convert to a finite confidence interval and these symptoms were excluded from the graph.

**Figure 8 healthcare-12-00945-f008:**
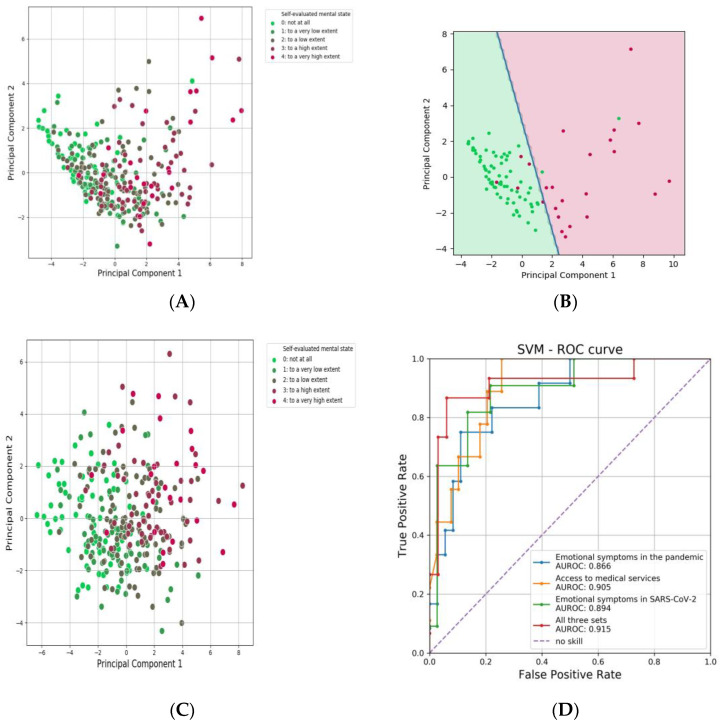
PCA analysis of the separation of responses by self-reported mental state. (**A**) dot plot of the separation of levels of psychological impairment based on affective symptoms during pregnancy, during illness, and access to medical services; (**B**) dot plot of the separation of levels of mental impairment based on all questions in the questionnaire; (**C**) the cutoff between no impairment and highly impaired mental status according to the SVM analysis based on the three sets of questions from (**A**); (**D**) the result of the logistic regression for the discrimination of patients with “very much” versus “not at all” impairment based on the three sets of questions.

**Table 1 healthcare-12-00945-t001:** Demographic information.

	Age		Medical Disorders Associated		Primiparous	Multiparous
317 women	18–25 years old	7%	Yes	13%	50%	50%
	26–25 years old	79%				
			No	87%		
	36–45 years old	14%				

**Table 2 healthcare-12-00945-t002:** General information about our participants during SARS-CoV-2 infection.

Status of Pregnancy			Difficult Access to Medical Services		Form of the Disease			
1st Trimester	2nd Trimester	3rd Trimester	Yes	No	Asymptomatic	Mild	Average	Severe
27%	42%	31%	25%	75%	7%	63%	28%	2%

**Table 3 healthcare-12-00945-t003:** After COVID-19 disease.

Recovery		Physical Condition		Mental State	
Very easy	25%	Not at all	20%	Not at all	22%
Easy	34%	To a very small extent	20%	To a very small extent	22%
Neither hard nor easy	27%	To a small extent	34%	To a small extent	31%
Hard	11%	To a large extent	22%	To a large extent	17%
Very hard	2%	To a very large extent	4%	To a very large extent	8%

## Data Availability

Data are contained within the article.

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
