# Peer review of "The Psychological Impact on Romanian Women Infected with SARS-CoV-2 during Pregnancy"

_healthcare, 2024, doi:10.3390/healthcare12090945_

Round 1

Reviewer 1 Report

Comments and Suggestions for Authors

I would thank the authors for this interesting manuscript.

After reviewing I found some errors that has affected the quality of the manuscript.

Major comments:

The main concern is related to the quality of the presentation of the results. In fact, the authors tend to study each factors separately (with eventually its figure) (this is an article not a thesis and we should be as concise as possible). Thus, I recommend to associate the figures and tables that one can follow the sequencing of the ideas:

You can associate Figure 5, 6 and 7 in one table (the the same for figures 9, 10 and 11. Also you can make the figure 13 in one table (we cannot associate a table with a figure)).

You should also delete figure 1 and associate it with the other "missed" demographic characteristic in another table (19% of the participants were aged between 26 and  35 years: where are the others? Where can we find them? Are there other parameters? (month or trimester (it shold be described here? Primiparous o multiparous….

You can also delete the response "no" from figure 2, 3, 4 and 8 to have uniform bars (we know that the total is 100%). You can also reduce the size of the figure to reduce the length of the manuscript.

You should also delete the title from the "kept" figures (you have two titles) and standardize all their titles (i.e: figure 3 ?).

Figure 12 separate between the title, the description and the text.

Introduction:

The introduction contains multiple self statement, and non referenced paragraphs. Also, the question (he hypothesis) of the study is not clear. What was done before in the country and around the world? What is the need of the study. Also, the ideas are not well organized and not correlated, readers tend to understand that there multiple "separate" paragraphs (see also my comments below).

At last, the manuscript should undergo an English language editing

Minor revisions:

Standardize the use of SARS-CoV-2 (not SARS-CoV2…) and COVID-19 (not COVID, covid-19, the disease of COVID-19...) throughout the manuscript.

You should also standardize the use of these two terms, COVID-19 or SARS-CoV-2  infection  

Revise the title

You should complete you abstract with the quantitative results . it is mainly based on qualitative descriptions.

Line 21: " Pregnant women who contracted SARS-CoV-2 infection faced much more negative feelings". Ou begin our results with a conclusion. Also, this statement has no relation with your work. Have you studied non pregnant women (why much more? ). Revise

Introduction:

Line 35: delete "after the first cases o COVID-19".

Add referenes for lines 39, 42, 62

Line 50: "generally…isn't associated with severe…". This stamen has no sense. The disease could be severe (and also deadly) for healthy individuals, how can pregnant women escape this rule. Revise  

Line 56: delete " more data…reports"  

Line 62-63: why did you provide this information (post-partum)?

Line 69-71: "in our opinion…" delete

Methods:

Line 79: add the references used to prepare the questionnaire

Line 115: what was the p value?

Results:

You should begin with the total number of participants

Line 126: the main source of information….about what?

Figures: the sentences and words should begin with capital cases

You should revise all the paragraphs of the results (for English language)

Line 209: revise " to with"

Discussion:

You should revise the discussion: the most important findings of the study were discussed in the last paragraph (only three paragraphs: 283-285) while the first part "lines 229-282" seem to be another introduction. Revise and discuss you results.

Conclusion :

Delete "much more".

Comments on the Quality of English Language

Extensive language editing required

Author Response

Dear Reviewer,

Please find attached our responses to your suggestions and thank you for your time and indications.

Reviewer 2 Report

Comments and Suggestions for Authors

Dear Authors,

It has been a pleasure to review this manuscript which aims to assess the maternal psychological impact on Romanian women infected with SARS-CoV-2 during pregnancy, their concerns and to determine the best measures to limit the negative outcome.

For the sole purpose of improving the quality of this manuscript, I would like to make several comments:

This is a well-written document with a clear objective and methodology.

I recommend changes to the Introduction to make it a coherent justification of the research problem.

The research questions and the hypotheses derived from them are missing. Please complete it.

Adequate discussion is generally missing. The authors compare their results with others but do not discuss the possible reasons or implications of their findings. I would recommend incorporating a section on the limitations of the study.

Author Response

(The authors gave the same response as above.)
